# The Solvent Dimethyl Sulfoxide Affects Physiology, Transcriptome and Secondary Metabolism of *Aspergillus flavus*

**DOI:** 10.3390/jof7121055

**Published:** 2021-12-09

**Authors:** Laura H. Costes, Yannick Lippi, Claire Naylies, Emilien L. Jamin, Clémence Genthon, Sylviane Bailly, Isabelle P. Oswald, Jean-Denis Bailly, Olivier Puel

**Affiliations:** 1TOXALIM (Research Center in Food Toxicology), Université de Toulouse, INRAE, ENVT, EI-Purpan, Toulouse 31027, France; laura.hm.costes@gmail.com (L.H.C.); yannick.lippi@inrae.fr (Y.L.); claire.naylies@inrae.fr (C.N.); emilien.jamin@inrae.fr (E.L.J.); sylvianebailly7@gmail.com (S.B.); isabelle.oswald@inrae.fr (I.P.O.); olivier.puel@inrae.fr (O.P.); 2Metatoul-AXIOM Platform, MetaboHUB, National Infrastructure for Metabolomics and Fluxomics, Toulouse 31000, France; 3INRAE, US1426, GeT-PlaGe, Genotoul, 31326 Castanet-Tolosan, France; clemence.genthon@inrae.fr

**Keywords:** dimethyl sulfoxide, DMSO, *Aspergillus flavus*, RNA-seq, aflatoxins, ustiloxin B, cyclopiazonic acid, secondary metabolism

## Abstract

Dimethyl sulfoxide (DSMO) is a simple molecule widely used because of its great solvating ability, but this solvent also has little-known biological effects, especially on fungi. *Aspergillus flavus* is a notorious pathogenic fungus which may contaminate a large variety of crops worldwide by producing aflatoxins, endangering at the same time food safety and international trade. The aim of this study was to characterize the effect of DMSO on *A. flavus* including developmental parameters such as germination and sporulation, as well as its transcriptome profile using high-throughput RNA-sequencing assay and its impact on secondary metabolism (SM). After DMSO exposure, *A. flavus* displayed depigmented conidia in a dose-dependent manner. The four-day exposition of cultures to two doses of DMSO, chosen on the basis of depigmentation intensity (35 mM “low” and 282 mM “high”), led to no significant impact on fungal growth, germination or sporulation. However, transcriptomic data analysis showed that 4891 genes were differentially regulated in response to DMSO (46% of studied transcripts). A total of 4650 genes were specifically regulated in response to the highest dose of DMSO, while only 19 genes were modulated upon exposure to the lowest dose. Secondary metabolites clusters genes were widely affected by the DMSO, with 91% of clusters impacted at the highest dose. Among these, aflatoxins, cyclopiazonic acid and ustiloxin B clusters were totally under-expressed. The genes belonging to the AFB1 cluster were the most negatively modulated ones, the two doses leading to 63% and 100% inhibition of the AFB1 production, respectively. The SM analysis also showed the disappearance of ustiloxin B and a 10-fold reduction of cyclopiazonic acid level when *A. flavus* was treated by the higher DMSO dose. In conclusion, the present study showed that DMSO impacted widely *A. flavus*’ transcriptome, including secondary metabolism gene clusters with the aflatoxins at the head of down-regulated ones. The solvent also inhibits conidial pigmentation, which could illustrate common regulatory mechanisms between aflatoxins and fungal pigment pathways. Because of its effect on major metabolites synthesis, DMSO should not be used as solvent especially in studies testing anti-aflatoxinogenic compounds.

## 1. Introduction

Dimethyl sulfoxide ((CH_3_)2SO or Me2SO or DMSO) is a simple amphipathic molecule discovered in the late 19th century as a by-product of the paper making process [1]. DMSO is a manufactured solvent which is also a naturally occurring substance, reported at low concentrations (<0.05 ppm to 3.7 ppm) in diverse products such as tomato paste, black tea leaves, coffee, or raspberries [2]. This molecule gained notoriety for its ability to penetrate skin and other biological membranes. Due to its physico-chemical properties, this colorless liquid is widely employed as a polar aprotic solvent, which can dissolve a very large variety of polar and nonpolar compounds. All these properties explain its very large field of use, from cryobiology to toxicological research [3]. DMSO also has many pharmacological applications in both human and veterinarian medicines. In 1978, the FDA approved its human therapeutic indication for the treatment of interstitial cystitis, but studies are still on to develop other medical uses [4,5]. Since the 1960s, a large amount of in vivo and in vitro studies has been conducted to evaluate the biological properties of this active molecule [6]. Among them, its properties as radical scavenger and anti-inflammatory compound are quite interesting but still little documented. Several studies had also underlined its antimicrobial activity against *Pseudomonas aeruginosa*, *Microsporum canis* or *Candida* spp. for instance, which can alter minimum inhibition concentration results of antimicrobial drug tests when DMSO is used as vehicle solvent [7,8,9]. In the field of pathogenic fungal research, DMSO has already been employed as vehicle solvent for solubilizing water-insoluble compounds [10,11,12,13] or as oxidative stress modulator on *Aspergillus flavus* [14].

*A. flavus* is a famous saprophytic fungus which can contaminate crops worldwide in either field or post-harvest conditions [15]. It is considered as the most frequent source of aflatoxin B1 (AFB1), a potent carcinogenic mycotoxin, causing important agricultural and economical losses [16]. The hazardousness of this toxin has led the European Food Safety Authority to conduct a new risk assessment [17] and many countries to establish regulations setting the maximum levels allowed in many foods and feeds, AFB1 being the most strictly regulated [18,19,20]. Because of these public health and economic threats, *A. flavus* is one of the most widely studied fungal species, and the sequencing of the NRRL 3357 model strain genome has permitted to explore the fungal machinery, with a start focus on aflatoxin biosynthesis [21].

Sharma and Sharma [22] tested the effect of commonly used solvents (including DMSO, methanol, acetone and Tween-80) on *A. flavus*, *Aspergillus parasiticus*, *Aspergillus fumigatus* and *Aspergillus niger*. They reported changes in conidial aspect and germination as well as an inhibition of colony pigmentation in all fungi treated with DMSO. Its inhibitory effect on conidial pigmentation is long known in *A. niger* [23] and *A. flavus* [24], but no more recent studies have further investigated the biological effects of DMSO on fungi.

In this context, the aim of this study was to delineate the effect of the solvent DMSO on *A. flavus*. To this end, we combined an RNA-seq transcriptomic approach and an analysis of physiological responses to DMSO (mycelial growth, conidiation, germination). Our transcriptomic analysis showed a large number of regulated genes in response to DMSO associated with secondary metabolism, in particular aflatoxin biosynthesis, whose cluster genes were the most down-regulated. Due to its impact on several metabolites of interest, a particular attention has to be paid when using DMSO as a solvent.

## 2. Materials and Methods

### 2.1. Chemicals and Reagents

Mycotoxins used as standards for quantification were purchased from Sigma-Aldrich (Saint-Quentin-Fallavier, France). All solvents were analytical grade and purchased from Thermo Fisher Scientific (Illkirch, France).

### 2.2. Fungal Strain and Culture Conditions

*Aspergillus flavus* NRRL 62477 [25] used in this study was maintained in the dark on a Malt Extract Agar (MEA) medium (Biokar Diagnostics, Allone, France). 

To prepare the *A. flavus* inoculum, conidia from a seven-day culture at 27 °C were collected and suspended in Tween-80 (0.05%). The concentration of conidia in the suspension was determined using a hemocytometer. Plates were centrally inoculated with 1000 conidia then incubated for four days at 27 °C in the dark. Prior to inoculation, MEA was supplemented with chosen doses of DMSO. 

As a first step, six different concentrations of DMSO (8.8 to 281.6 mM) were tested to determine which doses to use for physiological and transcriptomic assays. The concentrations of 35.2 (D1) and 281.6 mM (D2) were thus selected according to their effect on conidial pigmentation (Figure 1). 

### 2.3. Physiological Assays

#### 2.3.1. Fungal Growth and Sporulation 

Fungal growth was evaluated by measuring colony diameters after four-day incubation.

For sporulation, four-day cultures were suspended in 40 mL of Tween-80 (0.05%) using a stomacher bag and spores were carefully scraped up off the mycelium. Samples were homogenized with a Stomacher Lab-Blender 400 for 90 s and filtered through sterile gauze. To recover the remaining spores, three supplementary rinses were performed, each one with 20 mL of Tween-80. Spores were then counted on a hemocytometer. Spore density was calculated as follows: SD = SC/(πr^2^) where SC is the spore count and r is the average colony radius. 

#### 2.3.2. Germination 

Spore germination was studied as described by Bluma et al. [26] and Gougouli and Koutsoumanis [27] with some modifications. Agar pieces (10 × 10 mm) were aseptically harvested from MEA plates amended or not with the highest dose of DMSO and transferred to microscope slides (done in triplicates). Slides have been then inoculated with 10^4^ spores (10 µL) applied on each agar pieces and incubated at 27 °C. After 4, 6, 8, 10 and 12 h at 27 °C, one slide of each condition was observed under microscope (×400) to evaluate germination rate and a potential delay to germination. Several microscope fields (containing an average of 35 ± 8 spores) were captured and further analyzed. Around 100 single spores per agar piece (300 spores per treatment) were examined. Spores were considered germinated when the germ-tube was longer than the spore diameter. The germination rate was calculated as TG (%) = (number of germinated spores/total spores) × 100.

Germination capacity was evaluated by plating sixty spores of *A. flavus* on MEA petri-dishes (6 replicates). The colony forming units (CFU) were then counted after 3 days of incubation at 27 °C.

#### 2.3.3. Spore Resistance 

Protocol for assaying conidial resistance was adapted from Levdansky et al. [28]. A conidia suspension of *A. flavus* (5 × 10^4^/mL) was transferred in 2 mL tubes containing ceramic beads 1.4 mm in diameter (Bertin, Montigny-le-Bretonneux, France), in triplicate. These tubes were then vortexed at 2500 rpm for up to 40 min. At each time point, a part of the suspension was collected, diluted to the hundredth (5 × 10^2^ conidia/mL), and 100 µL were plated on MEA plates in three technical replicates. The CFU were counted after 3 days of incubation at 27 °C, and survival rate (SR) was calculated as follows: SR = [(number of CFU at time X)/(number of CFU at time zero)] × 100.

### 2.4. Transcriptomic Study

#### 2.4.1. RNA Extraction 

RNA extraction was performed as described previously [29,30,31]. Cultures dedicated to RNA isolation were incubated in 6 replicates per condition (control, DMSO D1 and DMSO D2). Before inoculation, media were covered with sterile cellophane layers. After 4-day incubation at 27 °C, mycelia were separated from the medium and grounded in liquid nitrogen with mortar and pestle. Total RNA was then extracted from 100 mg of mycelium using a Qiagen RNeasy PlusMinikit (Qiagen, Hilden, Germany) with an additional QIAshredder homogenizer column (Qiagen) as recommended by the manufacturer. The concentrations and integrity of obtained RNA were analyzed using a Dropsense96 (Trinan, Gentbrugge, Belgium) and 2100 Bioanalyzer (Agilent Technologies, Santa Clara, CA, USA), respectively.

#### 2.4.2. RNA-Sequencing

A total of 18 samples (three conditions, six biological replicates per condition) were sequenced for this study. Two micrograms of total RNA was used to build sequencing libraries using Illumina TruSeq Stranded mRNA Library Prep Kit (Illumina, San Diego, CA, USA). Briefly, mRNAs were purified using poly-T magnetic beads then chemically fragmented (around 300 bp) under 94 °C. First and second strand cDNA synthesis were consecutively performed according to the manufacturer’s protocol. After that, cDNA fragments were adenylated on 3′ ends, and Illumina indexed adapters were ligated on both ends. Specific tagged cDNA fragments were then enriched in a PCR program including a denaturation step at 98 °C (30 s) followed by 10 cycles of amplification (98 °C for 10 s, 60 °C for 30 s and 72 °C for 30 s) and a final 30 s step at 72 °C. Finally, pooled libraries were sequenced on one lane of Illumina HiSeq 3000 platform, resulting in 150 pb paired-end reads. Transcriptomic data and experimental details are available in NCBI’s Gene Expression Omnibus [32] and are accessible through GEO Series accession number GSE189211 (https://www.ncbi.nlm.nih.gov/geo/query/acc.cgi?acc=GSE189211).

#### 2.4.3. Bioinformatic and Biostatistic Data Analysis

Raw pair-ended generated reads were first trimmed to remove Illumina adapters and low-quality reads using Trim_Galore version 0.4.0 (https://github.com/FelixKrueger/TrimGalore, accessed on 8 December 2021) [33].

Trimmed reads were then aligned to the *A. flavus* NRRL 3357 genome using STAR software version 2.5.0b in a 2-pass mode. Then, FeatureCounts version 1.4.5-p1 [34] was applied to perform read summarization at the gene level, generating a raw count matrix. The statistical analysis was conducted using R [35]. Count data matrix was filtered to select genes with count per million reads (cpm) ≥ 1 in at least 5 samples among 18 and genes with at least 1 read in at least 5 samples per condition in at least one condition. The dispersion estimation and differential analysis was performed using edgeR package [36]. A correction for multiple testing was applied using Benjamini–Hochberg procedure [37] for the control of the False Discovery Rate (FDR). Genes with *q*-value < 0.05 (FDR < 5%) were considered to be differentially expressed genes (DEG) between conditions. Hierarchical clustering was applied to the samples and the differentially expressed genes using Pearson correlation coefficient of log cpm data as distance, and Ward’s criterion for agglomeration. The clustering results are illustrated as a heatmap of expression signals.

#### 2.4.4. Functional Analysis 

Gene ontology (GO) analysis and terms enrichment were conducted using FungiDB [38]. GO terms were considered enriched if their adjusted *p*-value was less than 0.05 after the Benjamini–Hochberg correction. The redundant GO terms were then summarized using REVIGO [39] and GO categories containing only one gene were removed.

#### 2.4.5. Expression Analysis of Secondary Metabolites Clusters

The annotation of secondary metabolites clusters of *A. flavus* was obtained from a consolidation between two sources: (1) the SMURF-predicted gene clusters from Giorgianna et al. [40] and Ehrlich and Mack [41] and (2) gene clusters which have been experimentally characterized, such as kojic acid, ustiloxin B, asperipin 2a or aspergillic acid [42,43,44,45,46,47] and summarized by Uka et al. [48].

For each secondary metabolites cluster, the percentage of DEG was calculated by dividing the number of DEG by the total number of detected genes. The regulation intensity (log2 Fold Change) was summarized at the cluster level using the mean log2 Fold Changes of the DEG. Clusters containing both up-regulated and down-regulated genes were considered to have a mixed regulation response and their log2 Fold Changes were transformed to absolute values before being averaged.

### 2.5. Secondary Metabolites Production Analysis

#### 2.5.1. Aflatoxin B1 by HPLC-FLD

Quantitative determination of aflatoxin B1 production by HPLC-FLD was adapted from Caceres et al. [31]. Culture media were first mixed with 30 mL of chloroform and agitated for 2 h on a horizontal shaking table (150 rpm at room temperature). Chloroform extracts were then filtered through a Whatman 1PS phase separator (GE Healthcare Life Sciences, Vélizy-Villacoublay, France) and evaporated to dryness at 60 °C. The residue was dissolved in a water–acetonitrile–methanol (65:17.5:17.5; *v*:*v*:*v*) mixture, then filtered (0.45 µm PTFE disks). Ten microliters were finally injected into the HPLC system, a Dionex Ultimate 3000 UHPLC system (Thermo Scientific, Illkirch, France) and analyzed using a LC column Luna^®^ C18 (125 × 2 mm, 5 μm, 100 Å) (Phenomenex, Torrance, CA, USA) at 30 °C. The eluent flow was 0.2 mL/min in an isocratic program with eluent A: eluent B (82.5%:17.5%) where eluent A is a mixture of acidified water (0.2% of acetic acid): acetonitrile (79:21; *v*:*v*) and eluent B is pure methanol. AFB1 was detected by FLD at 365 nm/430 nm as excitation and emission wavelengths, respectively. AFB1 peaks were confirmed using a coupled diode array detector (DAD) and sample concentrations were calculated based on a linear calibration curve of standards.

#### 2.5.2. Other Studied Secondary Metabolites by Mass Spectrometry

Analysis of other metabolites production by *Aspergillus flavus* was carried out on a HPLC coupled with a LTQ Orbitrap XL High-Resolution Mass Spectrometer (HRMS) (Thermo Fisher Scientific, Illkirch, France). 

Cultures dedicated to metabolites analysis were prepared and incubated according to the same protocol as for the transcriptomic assay (sterile cellophane layer, 1000 conidia inoculation, 4-day incubation at 27 °C and 6 replicates per condition). After the incubation period, 20 mL of ethyl acetate was added on mycelia previously separated from medium peeling cellophane layers. Samples were then shaken overnight, filtered through a Whatman 1PS phase separator (GE Healthcare Life Sciences, Vélizy-Villacoublay, France) and 15 mL of these extracts were finally evaporated to dryness at 60 °C. Residue was then dissolved in a mixture of water: acetonitrile (50:50; *v*:*v*), centrifuged at 10,000 rpm during 10 min, diluted to one-tenth and injected in the HPLC-HRMS system. A hypersil Gold C18 column (100 × 2.1 mm, 1.9 µm) from Thermo Scientific (Les Ulis, France) was used with a gradient program of H_2_O/CH_3_OH/CH_3_CO_2_H 95/5/0.1 (*v*:*v*:*v*) and CH_3_OH/CH_3_CO_2_H 100/0.1 (*v*:*v*), at a flow rate of 0.3 mL/min, at 40 °C. The following parameters of electrospray were applied: capillary voltage 4 kV, capillary temperature 300 °C, sheath gas flow rate 55 a.u., auxiliary gas flow rate 10 a.u. and tube lens offset −200 V in the negative mode; capillary voltage 4 kV, capillary temperature 300 °C, sheath gas flow rate 30 a.u., auxiliary gas flow rate 10 a.u. and tube lens offset 120 V in the positive mode. High resolution mass spectra were acquired between *m*/*z* 80 and 1300, in the centroid mode at a resolution of 30000 (*m*/*z* 400). Raw data were converted in mzXML format using Proteowizard and CWT. Then data were processed using XCMS offline software with centwave algorithm (ppm = 10, peakwidth = (10–70), snthresh = 10, bw = 5, mzwid = 0.01). All data analysis steps were performed using the collaborative portal workflow4metabolomics. The identity of detected secondary metabolites was confirmed by tandem mass spectrometry experiments (MS/MS) achieved on the same instrument.

### 2.6. Statistical Analysis 

All data are presented as the means ± Standard Error of the Mean (SEM). Data were analyzed using GraphPad Prism v4 software. For assays conducted in triplicate (germination rate and spore resistance), Mann–Whitney test was used to test for the differences between control and treated samples. Fungal growth, sporulation and germination capacity assays were carried out using six replicates, and the variations between control and treated groups were tested using Student’s *t*-test. Differences were considered to be significant when the *p*-value was lower than 0.05. 

## 3. Results

### 3.1. Dose-Dependent Effect of DMSO on Pigmentation

The presence of DMSO in the culture medium resulted in a dose-dependent depigmentation of *A. flavus* colonies, the color of which evolved from a typical yellow-green color (control) to creamy-white for the highest tested dose 282 mM (Figure 1). 

For subsequent assays, the doses of 35 mM (D1) and 282 mM (D2) were selected on the basis of pigmentation inhibition intensity. D2 allowed a complete depigmentation of culture whereas D1 corresponded to an intermediate active concentration with a mild effect on pigmentation, the greenish color being still visually detectable. 

Additional experiments indicated that modifications observed in response to DMSO were reversible. The green pigmentation was reestablished after cultivating treated spores on DMSO-free MEA.

### 3.2. Effect of DMSO on Physiological Parameters (Fungal Growth, Sporulation, Germination and Spore Resistance)

As illustrated in Figure 2, fungal growth was not significantly impacted by the lowest dose D1 and showed a limited modulation (8.9% decrease of colony diameter) at the highest DMSO concentration D2.

The influence of DMSO on sporulation was first investigated through the count of conidia (Figure 3A). A dose-dependent decrease of the total spore count was observed in the presence of DMSO with a reduction of 19.2% (*p* < 0.05) and 30.8% (*p* < 0.001) in culture treated with low and high level of solvent, respectively. However, when the number of conidia was normalized to the surface of the fungal colony, this impact was less important; a significant decrease being only observed at D1 (16.6%, *p* < 0.05).

The impact of DMSO was then evaluated on the rate of germination up to 12 h of incubation (Figure 3B, top-graph). No significant difference for germination rate was observed comparing D1 to control samples (*p* > 0.05). Nevertheless, we observed a brief delay in germination for the treated conidia, without any observable consequence at the macroscopic level. We also estimated the effect of DMSO on the germination capacity (Figure 3B, bottom-graph) and the number of CFU after 3 days at 27 °C was higher for cultures grown on DMSO, the increase being only significant for D2 (+27.4%, *p* < 0.05).

As already mentioned, a total depigmentation of conidia was observed in response to DMSO. Because of the multiple roles of pigments in spore resistance, the resistance of control (green-pigmented) and treated (depigmented) conidia to disruption induced by mechanical treatment using was assessed with ceramic beads. As shown in Appendix A, no significant difference was observed between the two survival curves.

### 3.3. Transcriptomic Response of Aspergillus flavus to DMSO Treatment 

To go further and explore differences in *A. flavus* transcriptomes in response to DMSO, an RNA-seq analysis was conducted. An average of 21.8 million reads per library was obtained, which were then aligned to the *A. flavus* NRRL 3357 genome (Appendix A). The gene level counts summarization resulted in a count table for 13,485 genes which was filtered to exclude rarely detected genes (10,619 genes selected). 

Principal component analysis (PCA) performed on the log2cpm values of filtered dataset revealed a clear separation of the three groups (control, low and high dose of DMSO) along the first component which explained 58% of the total variance (Figure 4A). Control samples were clearly separated from DMSO-treated ones on this first axis. However, samples treated with the low dose of DMSO were closer to the control ones than to the samples treated with high dose.

The differential expression analysis conducted on filtered dataset revealed 4891 significantly differentially expressed genes (DEG) in response to DMSO (D1 vs. control and D2 vs. control). As illustrated in Figure 4B, 2396 and 2262 were specifically up- and downregulated, respectively, upon exposure to the highest dose of DMSO (D2), whereas only 7 and 20 genes were up- and down-regulated upon exposure to the lowest dose of DMSO (D1). There is no overall major unbalance in the repartition between up- and down-regulated genes under the two conditions.

The expression profiles of the 500 most significantly regulated genes (according to *q*-value) upon DMSO exposure are presented in a heatmap (Figure 5A). The clustering confirms the close expression pattern between control and D1 samples, whereas D2 samples displayed a distinct expression pattern. Regarding the left clustering dendrogram, nine gene clusters presenting specific expression profiles were identified (Figure 5B). A large majority of the most regulated genes showed a dose-dependent under-expression (cluster 2; 244 genes) or over-expression (clusters 7, 8 and 9; 279 genes), with a lower effect at D1. In addition to that, some genes present a more surprising biphasic expression profile in response to DMSO (clusters 3, 4 and 5; 34 genes) with a stronger regulation at D1 and a lower effect at D2. Finally, some genes (clusters 1 and 6; 58 genes) showed a similar regulation profile by both doses of DMSO.

### 3.4. Functional Analysis of Differentially Expressed Genes

Gene ontology enrichment analysis was performed on DEG obtained in response to low (*n* = 241) and high (*n* = 4872) doses of DMSO, regardless the nature of their regulation (Appendix A). For each dose, the three GO domains: Biological Process, Molecular Function and Cellular Component were analyzed (Figure 6). 

For the GO domain “Biological Process”, the two most significantly over-represented categories for DEG in response to both doses of DMSO were “organic heteropentacyclic compound biosynthetic process” and “toxin metabolic process”, with almost 100% of the genes down-regulated. Each of these categories included 23 and 27 DEGs in response to D1 and D2, respectively, which almost all belonged to the aflatoxins’ cluster. For D1, the next over-represented terms were “secondary metabolic process”, “carbohydrate derivative transport” and “single-organism biosynthetic process”. For D2, the other over-represented terms were completed by “divalent metal ion transport”, “small molecule catabolic process” and “polysaccharide metabolic process”. 

For the GO domain “Cellular Component domain”, the over–represented categories were associated with cell wall; “cell envelope” and “external encapsulating structure part” for D1 treatment; “fungal-type cell wall”, “external encapsulating structure” and “cell periphery” for D2 treatment.

Noticeably, all GO terms in Biological Process and Cellular Component were enriched with a larger proportion of down-regulated DEG than up-regulated ones, especially for D1 treatment which showed seven enriched top GO terms with 100% down-regulated DEG.

For the GO domain “Molecular Functions” D1 treatment was associated with catalytic activity (“serine-type carboxypeptidase activity”, “glucosidase activity”, “oxidoreductase activity and acting on peroxide as acceptor”), interactions with carbohydrates (“carbohydrate binding”) and “antioxidant activity”. D2 treatment was also associated with catalytic activity (“oxidoreductase activity, acting on the CH-NH_2_ group of donors”, “ligase activity, forming carbon-nitrogen bonds”, “ligase activity”), in addition to other functions such as “active transmembrane transporter activity” and “cofactor binding”.

### 3.5. Effect of DMSO Treatment on Secondary Metabolism

#### 3.5.1. DMSO Impact on All Secondary Metabolism Gene Clusters

We next investigated the impact of DMSO on expression of genes belonging to 56 described or predicted Biosynthetic Gene Clusters (BGC). Figure 7 and Appendix A show that DMSO had an effect on many clusters, with a dose-dependent modulation. In response to a low dose of DMSO (D1), 9 BGC (16%) were regulated (6 down- and 3 up-regulated). Cluster 54 (encoding for aflatoxins) was the most regulated BGC, presenting more than 80% of its genes regulated.

The high dose of DMSO (D2) had a stronger impact with 52 regulated BGC (93% of total studied BGC). Among modulated clusters, 13 (23%) were up-regulated, 15 (27%) were down-regulated and 24 (43%) showed a mixed modulation (both up- and down-regulated genes). The two clusters which displayed the highest regulation at D1 (cluster 54 and cluster 3) also showed the strongest effect at D2. Cluster 54 was the most down-regulated BGC (log2FC −7.22); all other down-modulated BGC exhibited fold changes below 6 (log2FC <2.5), among which clusters 55 and 31 were the most impacted. On the opposite side, clusters 5 and 51 were up-regulated at both DMSO concentrations.

#### 3.5.2. DMSO Impact on Characterized Secondary Metabolites (Cluster and Production)

The metabolites produced by most of the SMURF-predicted BGC in *A. flavus* are still unknown. To date, secondary metabolites have been experimentally assigned to only twelve BGC [42,43,44,45,46,47,49,50,51,52]. Among these identified BGC, fold change details of those presenting almost all DEG in response to DMSO are presented as follows. In parallel, the impact of DMSO on the production of corresponding secondary metabolites has also been evaluated. Before detailing mixed-modulated clusters, full down-regulated characterized clusters are presented, starting with the most down-repressed cluster in response to both doses of DMSO which is aflatoxins cluster (#54).

Aflatoxin B1

Biosynthesis of aflatoxins involves 30 genes among which *aflR* and *aflS* play a role in internal regulation of aflatoxin gene cluster. Their expression changes to both DMSO doses are presented in Figure 8A where *aflU* is absent (“not detected”) in relation with the inability of *A. flavus* species to produce G-type aflatoxins. After DMSO exposure, all genes of the cluster were down-regulated. In response to low dose of solvent (D1), the down-regulation was not significant for *aflF*, *aflT* and *aflR* (*q*-values = 0.802, 0.084 and 0.064, respectively). The fold change decrease for the other genes was limited and ranged from −1.96 for *aflY* to −1.13 for *aflF*.

A high dose of DMSO (D2) led to a greater decrease in the expression of genes. Except for *aflT* and *aflF* (log2FC < 1), all the other genes in the cluster were more heavily repressed. The inhibition ranged from 5.88 (*aflA*) to 428.8 (*nadA*) folds. 

The two genes encoding regulators AFLR and AFLS appeared among the less affected genes and their repression degrees were similar at both doses (−1.33 and −1.34 at D1, −8.7 and −7.6 at D2, respectively) even if *aflR* did not show a significant regulation at D1.

The exposition of *A. flavus* cultures to D1 and D2 led to 63% and 100% inhibition of the AFB1 production, respectively (Figure 8B). It is interesting to note that the level of gene down-regulation was correlated with the intensity of AFB1 inhibition. The HPLC-HRMS analyses also showed a decreased production of several aflatoxin B1 intermediates (versicolorins A and B, O-methylsterigmatocystin) (Appendix A).

Cyclopiazonic Acid

Located near the aflatoxins cluster, cyclopiazonic acid (CPA) cluster (#55) also appeared to be under-expressed after DMSO treatment (Figure 9A). All four genes of the cluster were significantly down-regulated in response to the high dose of DMSO only, with fold changes from −3.17 for the MFS transporter (AFLA_139460) to 5.78 for the hybrid backbone enzyme (AFLA_139490/pks-nrps).

As observed for AFB1, CPA production was also decreased in response to DMSO in a dose-dependent manner, but CPA levels were still detectable at D2, with 33% and 86% of production inhibition at D1 and D2, respectively (Figure 9B).

Ustiloxin B

Ustiloxin B genes cluster (#31) also presented a global down-modulation after DMSO treatment. Only four genes were significantly under-expressed in response to low dose of solvent (*ustC*, *ustP1*, *ustD*, *ustF2*), whereas all genes showed a significant down-regulation after exposure of the highest dose (Figure 10). The observed fold changes for the cluster in response to D2 ranged from −1.45 (*ustT*) to −8.37 (*ustA*, precursor peptide). *ustR* (AFLA_095080 and AFLA_095090), which encodes for the regulator of the cluster, a fungal-type Zn(II)2Cys6 transcription factor, was significantly repressed at D2 (FC −1.93). To evaluate the impact of these modifications on metabolite production, the culture medium and mycelium were separately extracted. A metabolite sharing the same retention time and the same MS/MS spectrum with an ustiloxin B authentic standard was observed only in fungal extracts in HPLC-HRMS analyses in the positive ionization mode (Appendix A and Appendix A). A significant decrease and a disappearance of ustiloxin B from extracts were confirmed at D1 and D2 doses, respectively. 

Mixed regulated BGC: leporins, ditryptophenaline and kojic acid

Other identified clusters with 100% detected genes in our study displayed mixed regulation (Appendix A). For leporins genes cluster (#23), 3 (*orf4*, *orf5* and *orf6*) and 7 (*lepA*, *lepC*, *lepD*, *lepF*, *lepG*, *lepH* and *lepI*) out of 14 genes were significantly under-expressed and over-expressed, respectively, at high dose of DMSO. The gene encoding leporins cluster transcription factor *lepE* did not show significant changes in expression levels.

Ditryptophenaline (#4) and kojic acid (#56) clusters, both composed of three genes, displayed a slight and contrasted modulation of their gene expression, with only one gene significantly down-regulated for each cluster: *dtpA* for ditryptophenaline cluster (−1.19) and *kojT* for kojic acid cluster (FC −1.21). No impact of these weak down-regulations was observed on the production of both metabolites.

Fold change details of characterized clusters presenting more than one undetected genes (for example aflavarin or aspergillic acid) are available in Supplementary Data (Appendix A).

### 3.6. Impact of DMSO Treatment on Expression of Genes Related to Fungal Development

As DMSO modulates fungal development, it was important to analyze its impact on genes known to be involved in this function. Likewise, global transcription factors are known to coordinate both primary and secondary metabolism. The expression levels of several global regulators and genes related to conidiogenesis were significantly impacted after DMSO exposure (Table 1). As already noted, the DMSO dose D1 had lower impact than the highest dose, by over-expressing only two development-related genes: *veA* and *pksP*.

#### 3.6.1. Velvet Complex Members and Other Global Transcription Factors

*VeA* (AFLA_066460) coding for a global regulator gene belonging to the velvet complex, was over-expressed at the two doses, with a 1.24 and a 1.77-fold change compared to the control. As velvet family proteins, VosA and LaeA can interact with VeA to modulate various functions such as spore viability, secondary metabolism, sexual and asexual development. *vosA* (AFLA_026900) showed a significant over-expression in response to D2 only. Expression of *laeA* (AFLA_033290), a transcriptional factor known to regulate fungal secondary metabolism, was found down-expressed for D2 dose. Finally, the *mtfA* (AFLA_091490) gene presented a similar up-regulated response to *veA* (+1.26).

#### 3.6.2. Asexual Development and Pigmentation

Genes belonging to the so-called fluffy gene family (*fluG*, *flbA*-*D*) showed variable regulations. *FluG* (AFLA_101920) and *flbA* (AFLA_134030) were up-regulated in response to D2, whereas *flbB* (AFLA_131490), *flbC* (AFLA_137320) and *flbD* (AFLA_080170) were down-regulated. 

Noticeably, except *rodB* (AFLA_014260) which encodes a hydrophobin involved in cell wall integrity, some conidial specific genes (*conF* (AFLA_044800), *conJ* (AFLA_083110), AFLA_044790) were mainly up-regulated, with rather high fold changes compared to the control (4.95, 4.53, 3.63, respectively). By contrast, transcription of genes related to conidial development presented contrasted changes after D2 treatment, with both over- (AFLA_052030 *wetA*,) and under-expressed genes (AFLA_029620 *abaA*, AFLA_082850 *brlA*). 

*A. flavus* cultures were totally depigmented in response to D2, and surprisingly genes related to pigmentation (AFLA_006170 *pksP*/*alb1*/*wA*, AFLA_006180 *arb2* and AFLA_075640 *ayg1*) were up-regulated, with fold changes of 5.03, 1.45 and 2.26, respectively.

## 4. Discussion

DMSO is a very widely used solvent, but its biological properties remain little known, and only few studies had recently reported its effects on fungi. Thus, we chose to explore the impact of DMSO at macroscopic and molecular levels on the notorious pathogenic mold *Aspergillus flavus*, through physiological parameters and high-throughput transcriptomic analysis.

First, we studied the impact of increasing concentrations of DMSO on colonies aspects and more specifically conidial pigmentation, which we used as a basis to select two working doses. Researchers were attracted very early by this main morphological impact of this molecule, namely the conidial depigmentation [23,24]. In our assay, the loss of pigmentation was present from the first tested concentration (17.6 mM) but better macroscopically detectable from 35.2 mM. These data are coherent with results from Bean et al. [24] where a first noticeable depigmentation of *A. flavus* conidia was detected at 2500 ppm (32 mM). Apart from its effect on conidial pigmentation, DMSO presented only a slight impact on other measured fungal growth parameters such as colony diameter, conidiation (spores/cm^2^) and germination delay. The limited effect of DMSO on fungal development was already observed by Tillman and Bean [54], who found that a large range of DMSO concentrations (0 to 50,000 ppm) did not affect mycelial growth or sporulation in *Fusarium roseum*. 

To explore the mechanisms of action of DMSO at molecular levels, RNA-seq emerged as a relevant analytical method that has already been used for mapping global transcriptional changes in mycotoxin producer fungi, including *A. flavus*, for example in response to versatile environmental conditions [55,56] or aflatoxinogenesis inhibitory compounds [11,57,58].

Results from our transcriptomic analysis showed a contrasting amount of DEG in response to the two concentrations of DMSO. Indeed, 241 and 4872 genes were differentially expressed in response to D1 and D2, respectively. The number of selected DEG depends on the compound’s effects on the fungi but also on the cut-off threshold chosen and applied to the data (analyzed annotated genes). In comparison with other *A. flavus* transcriptomic studies focusing on inhibitory compounds, our large number of DEG was not so surprising. In their study, Wang et al. [56] revealed 453 DEG (366 up-DEG and 87 down-DEG) in response to 3 µg/mL of resveratrol which induced a 47% decrease of AFB1 production. Using the same cut-off filter (abs(log2FC) > 1; *q*-value < 0.005), we obtained 20 and 929 DEG after D1 or D2 treatment, respectively. Furthermore, our amount of DEG were comparable with other studies which applied a similar cut-off point [57]. In comparison, RNA-seq analysis studying the impact of environmental parameters such as water activity or temperature obtained a great number of DEG despite a stricter cut-off point at abs(log2FC) > 2 used for some studies [55,56,59].

The GO enrichment analysis of the DEG between untreated and DMSO-treated cultures showed an over-representation of terms related to secondary metabolism, and more particularly toxin metabolism for both doses. Indeed, our results showed that DMSO presented a major dose-dependent impact on secondary metabolic gene clusters in *A. flavus*. It is not always the same observation in other transcriptomic studies of *A. flavus*. For example, after resveratrol and 5-azacytidine exposure, the authors found that the expression of most clusters was not significantly impacted among the 55 predicted clusters of *A. flavus* [11,57]. In contrast, the number of regulated BGC seems higher in response to environmental parameters, such as water activity or temperature [59]. Here again, the results depend also on the chosen method to study the regulation of predicted BGC; for example, taking into account all genes or only backbone genes within each cluster, or the application of a cut-off threshold to regulated genes within the cluster. Beyond fungal secondary metabolism, quantitative and qualitative alterations of secondary metabolite production after DMSO exposure was previously reported on bacteria strains that exhibited enhanced antibiotic production [60].

Among all the studied BGC, aflatoxins cluster (#54) appeared to be the most down-regulated cluster in response to both doses of DMSO. The dose-dependent inhibition of AFB1 production by *A. flavus* is coupled with the down-regulation of all aflatoxin cluster genes with a higher impact observed with D2 compared to D1. A dose-dependent inhibition of AFB1 production by *A. flavus* in response to increasing doses of DMSO was already stated by Bean [61]. In our study, we observed a correlation between the level of gene expression regulation and the intensity of AFB1 inhibition following DMSO treatment. Although there was an absence of detectable AFB1 in the medium, the two regulators *aflR* and *aflS* were not the most repressed after D2 treatment. Actually, *aflR* and *aflS* transcription levels are not always significantly or highly significantly decreased between conducive and non-conducive conditions for AFB1 production [29,62]. In addition, an interaction between AflR and AflS has also been reported as necessary for aflatoxin biosynthesis, possibly forming an AflR-AflS activation complex which increases transcription of other cluster genes. Genes involved in first steps of AFB1 biosynthetic pathways (*aflA*, *aflB* and *aflC*, forming the polyketide structure) seemed less affected than the downstream ones in response to D2, as previously reported in response to eugenol treatment [29]. A possible interpretation is that the AflR/AflS complex was formed in few quantities, sufficient to activate transcription of first genes but not to initiate the expression of those involved in the later stages. This hypothesis could also explain the correlation between the level of gene down-regulation and the intensity of AFB1 inhibition in response to DMSO.

After DMSO exposure, cyclopiazonic acid displayed the same dose-dependent decrease regarding both production and mRNA levels of cluster genes (#55). This BGC, located directly near the AFB1 cluster, does not contain an internal transcriptional regulator such as *aflR* or *aflS*. Several observations suggested a link between productions of AFB1 and CPA, and *aflR* could act as a regulatory protein for both clusters, explaining their apparent co-regulation by DMSO [49,63].

We observed that DMSO exposure strongly affected the ustiloxin B production. Originally reported from the rice pathogenic fungus *Ustiloginoides virens*, ustiloxin B belongs to a ribosomally synthetized and post-translationally modified peptides (RiPPs) class produced by a wide range of microorganisms and widely distributed in the *Aspergillus* genus [64,65]. Identified in the *A. flavus* genome by MIDDAS-M, a motif independent algorithm that predicts the BGCs [66], ustiloxin B gene cluster contains 15 genes (AFLA_09940-AFLA_095110) [51,53]. Except *UstU* (AFLA_094970) not detected in all cultures, all ustiloxin B genes including *UstR* encoding a specific transcription factor, were strongly down-expressed when *A. flavus* was exposed to the higher DMSO dose. Another BGC similar to ustiloxin B BGC, but reduced to four genes (AFLA_041370-AFLA_041400), was identified in *A. flavus* as responsible for the biosynthesis of asperipin 2a, a novel cyclic RiPP [64]. No impact of DMSO treatment was observed on the expression of these genes. The asperipin 2a BGC contains no gene encoding peptidase, suggesting the involvement of peptidases coded by genes located outside of the gene cluster. Regarding the feature, our results showed there is no crosstalk between both biosynthesis pathways since (i) the asperipin 2a production is not affected by DMSO treatment and (ii) *UstP* gene coding to peptidase is strongly down-regulated when *A. flavus* is treated by the highest DMSO dose.

Many regulator genes, external to BGC, are known to be directly or indirectly involved in regulatory mechanisms of secondary metabolism. Among them, LaeA is a well-known global regulator of fungal secondary metabolism, conserved in numerous filamentous fungi [67]. Microarray analysis on *laeA* mutant of *A. flavus* (deletion or over-expression) revealed a positive regulation role of LaeA for 24 clusters out of 55 predicted by SMURF [15], which is relevant with the decreased mRNA levels of *laeA* found in response to the highest dose of DMSO in our study. Moreover, the recently described ustiloxin B cluster also seems regulated by LaeA [68].

Although the expression pattern of *laeA* is correlated with that of several BGC, this is not the case for the *veA* gene. VeA is a key global regulator, which is currently reported to be a positive regulator of secondary metabolites in *Aspergillus* species [69,70,71]. In our study, transcription levels of *veA* were increased after exposure to both concentrations of DMSO, whereas several BGC such as aflatoxins or cyclopiazonic acid were down-regulated. A such inverted expression pattern had already been observed by Spröte and Brakhage [72] who hypothesized a repressor role of VeA on penicillin biosynthesis. Another explanation could be the multiform regulation network in which VeA and LaeA are involved through the velvet complex, a heteromeric protein structure that coordinates a large range of biological functions [73].

Moreover, VeA governs and cooperates with a great number of other genetic elements. It can interact with mtfA, another conserved transcription factor involved in the regulation of fungal development and secondary metabolism [74]. Zhuang et al. [75] reported that *mtfA* over-expression led to a severe decrease of AFB1 production associated with a down-regulation of *aflR*, which is consistent with our results. In our study, *mtfA* and *veA* were both up-regulated in response to the highest concentration of DMSO. This concurring over-expression of *veA* and *mtfA* has already been observed in *A. flavus* in response to other anti-aflatoxinogenic compounds, such as eugenol or *Micromeria graeca* extract [29,30].

DMSO also induced a dose-dependent conidial depigmentation of *A. flavus*. Pigments and melanin produced by fungal species are related to many roles, such as protection, pathogenicity or conidial structuring [24,76,77]. To check the DMSO-linked depigmentation impact on this last-mentioned function, we compared the mechanic resistance between control-pigmented and DMSO-depigmented conidia after disruption by ceramic beads. According to this assay, the lack of pigmentation due to DMSO did not seem to weaken the conidial structure. 

However, the co-inhibition of both AFB1 production and conidial pigmentation observed in our study is of interest. Indeed, aflatoxins and fungal pigments are both synthetized via enzymes from the polyketide synthase family. Thus, Dzhavakhiya et al. [78] recently explored the hidden link between discoloration and inhibition of aflatoxinogenesis by studying several potential inhibitors of AFB1 biosynthesis through their ability to block the conidial pigmentation in *A. flavus*. They identified one compound, compactin, which seemed to block both pigmentation and toxin production. An anti-aflatoxinogenic concentration of compactin induced a fluffy white phenotype (aerial mycelium) along with a strong inhibition of spore formation, in contrast with DMSO which depigmented conidia without reducing sporulation itself. Following their hypothesis, and considering that production of aflatoxins is intimately related to conidiation [79], DMSO could act on a common regulatory mechanism involved in both pathways producing pigments and aflatoxins during conidiogenesis. The unexpected over-expression of pigmentation related genes in response to DMSO may be explained by a feedback regulation, siRNA type for example, to restore an essential fungal element.

Although no abnormalities were found in conidiogenesis, germination and spore resistance, other genes related to spore maturation, germination and resistance were over-expressed when *A. flavus* was treated with D2 DMSO. These included *conF* and *conJ,* the orthologs of *con-6* and *con-10* of *Neurospora crassa*. Regulated by WetA [80], they exhibit redundant functions in the control of germination and participate in the protection of spores against dryness [80]. In addition, regulated by WetA in *A. nidulans* [80,81], *dewA*, *cetL*, *cetJ* as well as the orthologous gene AN10040 (AFLA_099050) are over-expressed 2.58, 2.84, 3.72 and 4.34 times, respectively. Among the 44 most up-regulated genes (FC > 4), fourteen genes (Appendix A) have orthologous genes determined by Chip-seq as potential target gene for VosA [82]. The up-regulation of these genes can be explained by the over-expression of *vosA*.

Among the genes that were the most down-regulated by a higher dose of DMSO are *CnxE* and *CnxG*, involved in the last two steps of pyranopterin-based molybdenum factor biosynthesis [83,84]. This cofactor is a structural component of most molybdoenzymes involved in the reactions required for the carbon, nitrogen and sulfur cycles in nearly all organisms [85]. Interestingly, molybdenum salts have been reported to reduce aflatoxin B biosynthesis without affecting the growth of *A*. *flavus* [86]. This observation has not been supported by studies elucidating the molecular mechanisms of such inhibition. Little is known about the homeostasis of molybdenum, and the mechanisms that regulate the physiological optimal intracellular level of molybdenum are poorly understood. Therefore, the hypothesis of a second chelating function of pterin in order to maintain a level of molybdenum favorable to the proper functioning of the fungal cell is not ruled out.

## 5. Conclusions

To conclude, we demonstrated a noticeable impact of DMSO on both physiology and transcriptome of the pathogenic fungus *A. flavus*. The transcriptomic analysis revealed numerous DEG associated with secondary metabolism in general, and aflatoxin cluster genes in particular which appeared as the most down-regulated genes of the study at the highest DMSO dose. Numerous affected DEG were also involved in development functions, with a notable over-expression of pigmentation related genes after DMSO exposure despite completely depigmented colonies. Thus, the co-inhibition of both AFB1 production and conidial pigmentation appears interesting through the hypothesis of a common regulatory network shared between aflatoxins and fungal pigment pathways on which the DMSO could act. However, this complex link remains to be deeply explored in further studies.

The depigmented conidia resulting from DMSO treatment were observed in other toxinogenic fungi such as *Aspergillus niger* [23] or *Penicillium expansum* (personal communication). To enlarge our approach to study mycotoxinogenesis regulative mechanisms, the impact of DMSO could be tested on other fungal species producing other mycotoxins. A multi-species comparative transcriptomic analysis could unravel biological functions commonly regulated by DMSO.

Our study revealed a non-negligible effect of the widely used solvent DMSO on toxinogenic fungi, especially on secondary metabolism; these results should be carefully taken in consideration when DMSO is used as vehicle solvent, especially for solubilizing compounds to test their effect on fungal cultures. We observed a strong dose-dependent impact of DMSO on *A. flavus*. Depending on the species studied, it is therefore essential to plan to test the effect of such solvents at working concentrations, particularly on metabolites of interest that are monitored.

## Figures and Tables

**Figure 1 jof-07-01055-f001:**
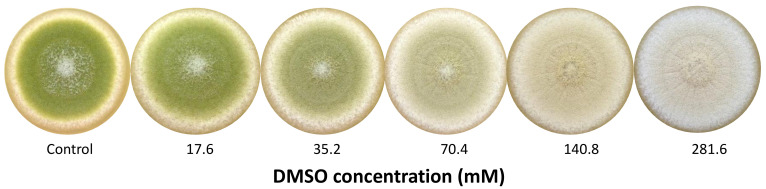
Macroscopic aspect of 8-day cultures of *Aspergillus flavus* NRRL 62477 in response to increasing dose of DMSO (mM).

**Figure 2 jof-07-01055-f002:**
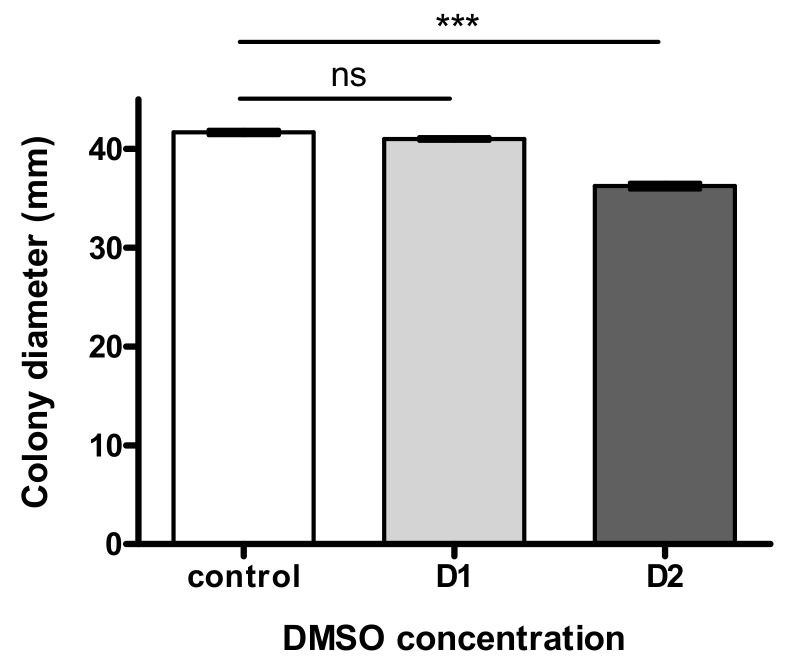
Effect of “low” (D1) and “high” (D2) concentration of DMSO on fungal growth of *Aspergillus flavus* NRRL 62477 after 4 days of incubation at 27 °C. ns = no significant difference; *** *p* < 0.001.

**Figure 3 jof-07-01055-f003:**
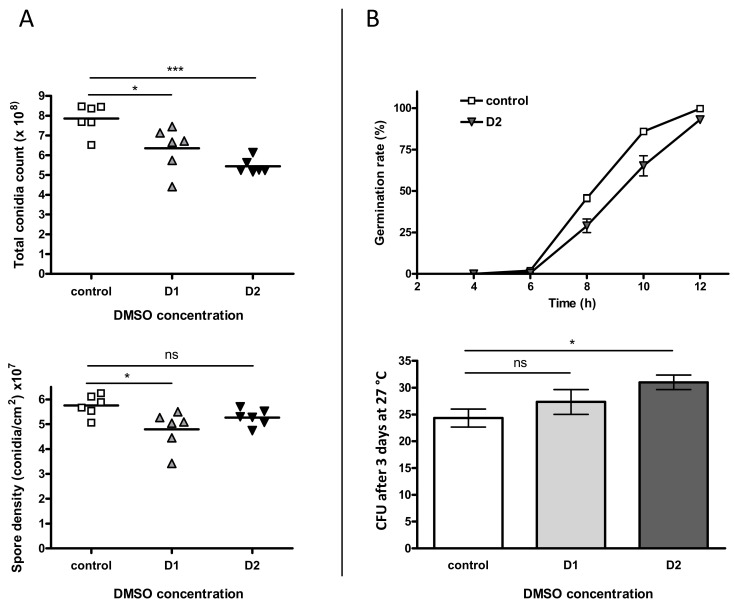
Impact of DMSO on both sporulation and germination of *Aspergillus flavus*. (**A**) The top-graph represents the total amount of spores in a 4-day colony after incubation at 27 °C in the dark. The bottom-graph shows the spore density (after normalization with colony area). In both graphs controls are represented with squares, D1-treated cultures with triangles and D2-treated cultures with inverted triangles. (**B**) Germination was studied at two scales: germination rate (top-graph) was examined microscopically for MEA (control) and MEA + 281.6 mM DMSO (D2). Germination capacity (bottom-graph) is corresponding to the CFU number after plating and incubating 60 theorical spores; ns = no significant difference; * *p* < 0.05 *** *p* < 0.001.

**Figure 4 jof-07-01055-f004:**
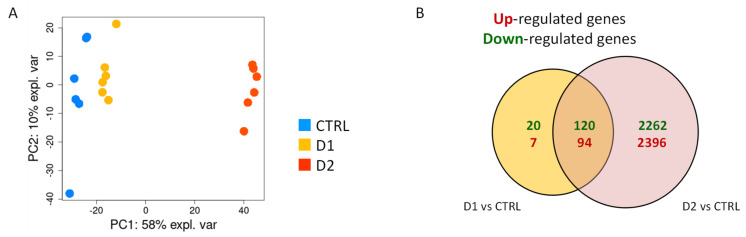
Transcriptomic impact of DMSO on *A. flavus* cultures. RNA-sequencing data were produced on 6 replicated cultures of *A. flavus* (control, D1 and D2 DMSO-treated). The data pre-treatment resulted in a count table of detected genes (10,619) for 18 samples. (**A**) Projection of samples on principal components 1 (PC1) and 2 (PC2) from a Principal component analysis (PCA) of log_2_cpm values. A differential analysis identified 4891 regulated genes (FDR < 5%) in response to D1 or D2. (**B**) Venn diagrams of differentially expressed genes (up- or down-regulated) in response to the two doses of DMSO.

**Figure 5 jof-07-01055-f005:**
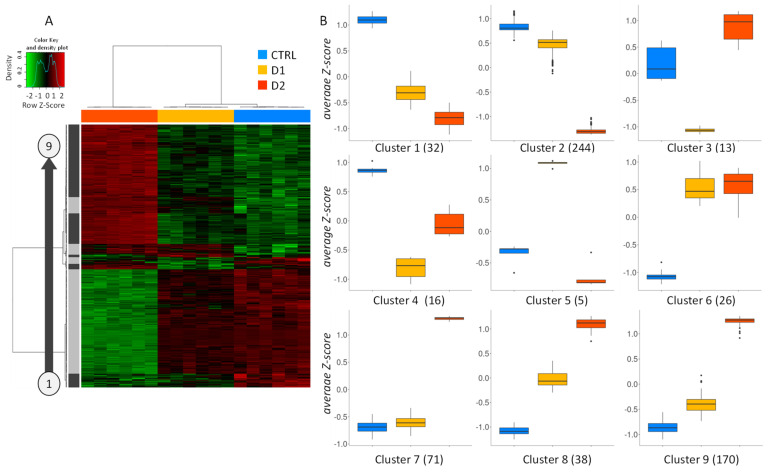
(**A**) Heatmap of the top 500 most significantly regulated genes in D1 vs. CTRL or D2 vs. CTRL comparisons. Red and green indicate values above and below the mean averaged centered and scaled log2cpm values (Z-score), respectively. Black indicates values close to the mean. According to the gene clustering (left panel), 9 gene clusters showed specific gene expression profiles. (**B**) Boxplots representing average Z-score of the 9 clusters (cluster 1 to cluster 9). Number of genes in each cluster is indicated in brackets.

**Figure 6 jof-07-01055-f006:**
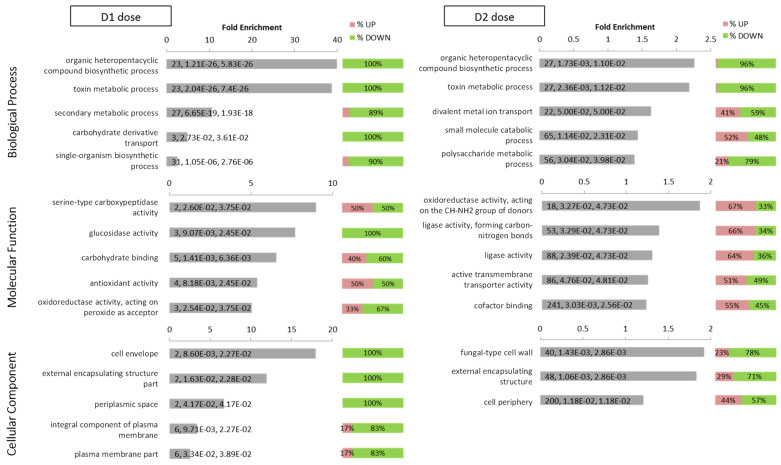
GO enrichment analysis of regulated genes by D1 and D2. Top 5 enriched GO categories according to the Fold Enrichment in Biological Process, Molecular Function and Cellular Component for each DMSO condition (D1 and D2). For each category, the label indicates the total number of regulated genes assigned to the specific GO category, the *p*-value and the adjusted *p*-value after Benjamini–Hochberg correction. The pink/green bars on the right of bar-plots show the percentage of up/down-regulated genes among each GO category.

**Figure 7 jof-07-01055-f007:**
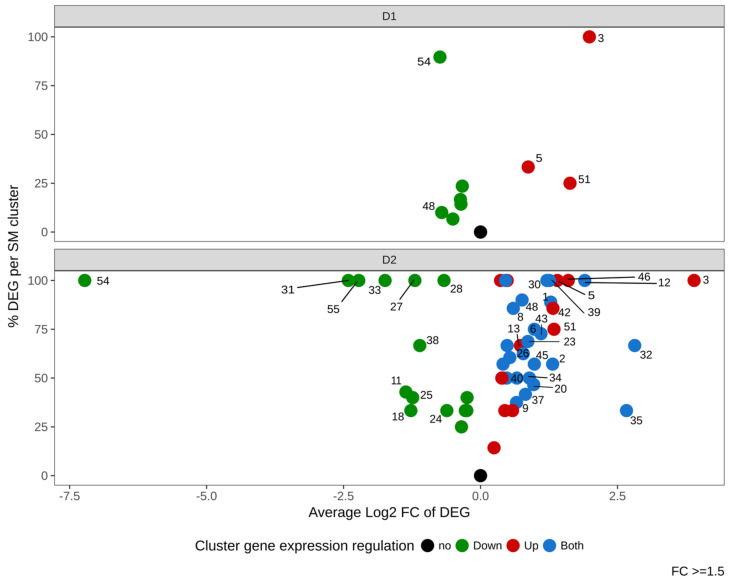
Summarized gene expression regulation of the 56 Secondary Metabolites Gene Clusters (SMC) of *Aspergillus flavus* in response to DMSO (D1 and D2). The x-axis represents the average gene expression log_2_ Fold Change in each SMC (“intensity of the regulation”). The y-axis shows the percentage of differentially expressed genes (DEG) within all genes of the SMC (nb DEG/nb detected genes). Green, red and blue dots indicate down-regulated, up-regulated and mixed-regulated (both up and down-regulated) clusters, respectively. The numbers of clusters with FC ≥ 1.5 (log_2_FC ≥ 0.58) are labeled next to the colored dots.

**Figure 8 jof-07-01055-f008:**
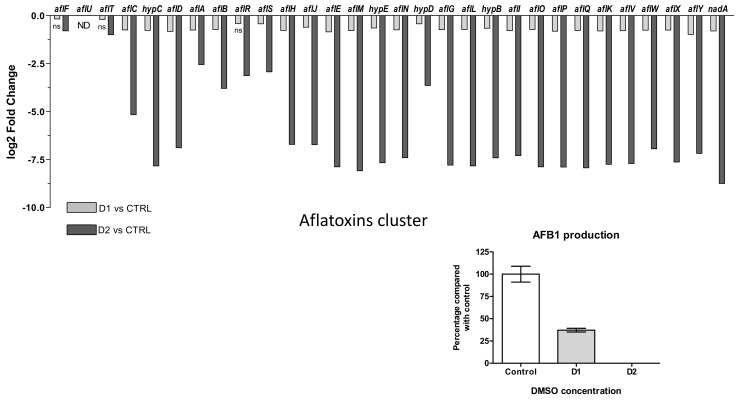
Effect of DMSO treatment on aflatoxin biosynthesis pathway. (**A**) Log_2_ Fold Change expression of genes belonging to the aflatoxin biosynthesis cluster (#54) (AFLA_139140-AFLA_139440) in response to DMSO (D1 and D2); only the genes with *q*-value > 0.05 are indicated as not significantly differentially expressed with “ns” below the bar-plot; “ND” means not detected. (**B**) Impact of DMSO on AFB1 production of *Aspergillus flavus* NRRL 62477 after 4 days at 27 °C. Results are expressed as percentage of the control value (+/−SEM).

**Figure 9 jof-07-01055-f009:**
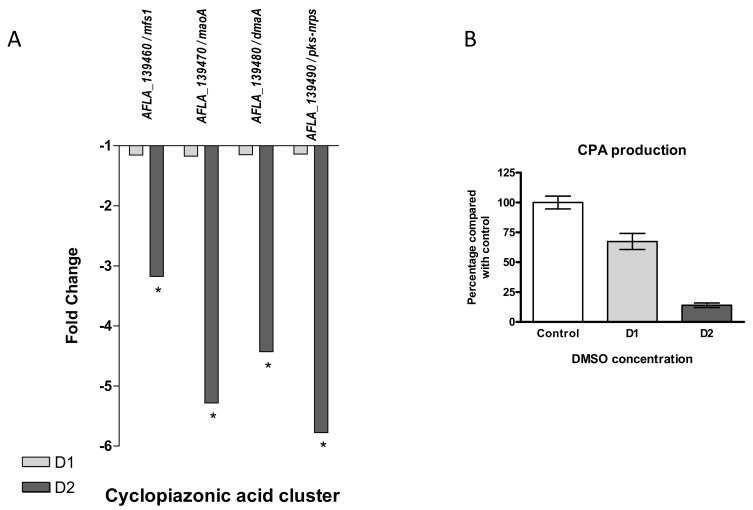
Effect of DMSO treatment on cyclopiazonic acid biosynthesis pathway. (**A**) Fold change expression of genes belonging to the cyclopiazonic acid cluster (#55) in response to DMSO (D1 and D2); * *q*-value < 0.05. (**B**) Impact of DMSO on CPA production of *Aspergillus flavus* NRRL 62477 after 4 days at 27 °C. Results are expressed as percentage of the control value (+/−SEM).

**Figure 10 jof-07-01055-f010:**
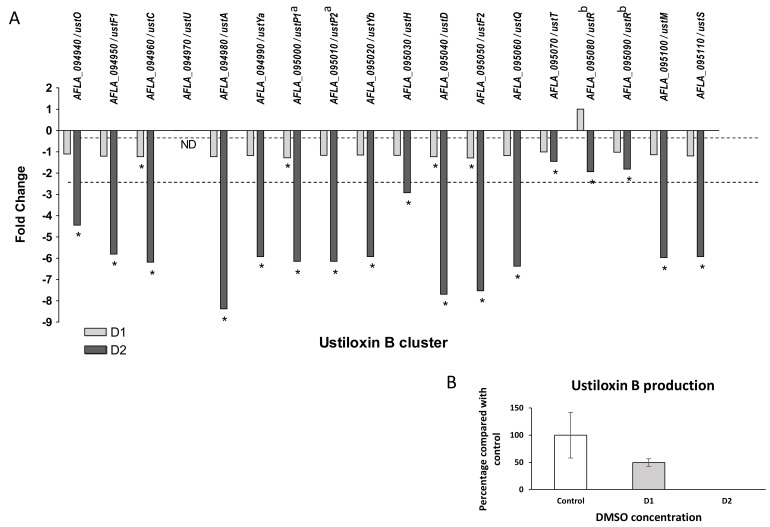
Effect of DMSO treatment on ustiloxin B biosynthesis pathway. (**A**) Fold change expression of genes belonging to the ustiloxin B cluster (#31) in response to DMSO (D1 and D2); * *q*-value < 0.05; ND = gene not detected; dotted line represents abs(FC) = 1 with no difference in expression against control samples. (**B**) Impact of DMSO on ustiloxin B production of *Aspergillus flavus* NRRL 62477 after 4 days at 27 °C. Results are expressed as percentage of the control value (+/−SEM). ^a^ AFLA_095000 and AFLA_095010 were concatenated to single gene under the Genbank accession number FAA01143 [53]. ^b^ AFLA_095080 and AFLA_095090 were concatenated to single gene under the Genbank accession number FAA01150 [53].

**Table 1 jof-07-01055-t001:** Differentially expressed genes related to the fungal development in *Aspergillus flavus*. Fold changes for under-expressed genes are marked in bold text.

Gene ID	Description	Common Name	Fold Change	*q*-Value	Fold Change	*q*-Value
AFLA_066460	developmental regulator AflYf/VeA	*veA*	1.24	6.75 × 10^−4^	1.77	2.98 × 10^−33^
AFLA_091490	C2H2 finger domain protein, putative	*mtfA*	-	-	1.26	4.30 × 10^−3^
AFLA_033290	regulator of secondary metabolism LaeA	*laeA*	-	-	−1.55	2.85 × 10^−12^
AFLA_101920	conserved hypothetical protein	*fluG*	-	-	1.24	6.83 × 10^−3^
AFLA_134030	extracellular developmental signal biosynthesis protein FluG	*flbA*	-	-	1.44	1.05 × 10^−5^
AFLA_131490	hypothetical protein	*flbB*	-	-	−1.20	1.77 × 10^−3^
AFLA_137320	C2H2 conidiation transcription factor FlbC	*flbC*	-	-	−1.62	1.96 × 10^−4^
AFLA_080170	MYB family conidiophore development protein FlbD, putative	*flbD*	-	-	−1.42	4.71v10^−5^
AFLA_082850	C2H2 type conidiation transcription factor BrlA	*brlA*	-	-	−1.14	4.76 × 10^−2^
AFLA_026900	developmental regulator VosA	*vosA*	-	-	2.21	9.16 × 10^−34^
AFLA_029620	transcription factor AbaA	*abaA*	-	-	−1.22	1.13 × 10^−2^
AFLA_052030	developmental regulatory protein WetA	*wetA*	-	-	1.49	1.93 × 10^−8^
AFLA_044790	conidiation-specific proteins		-	-	3.63	2.36 × 10^−16^
AFLA_044800	conidiation specific-protein Con-6	*ConF*	-	-	4.95	2.11 × 10^−42^
AFLA_083110	conidiation-specific protein Con-10	*ConJ*	-	-	4.53	4.08 × 10^−37^
AFLA_014260	conidial hydrophobin RodB/HypB	*rodB*	-	-	−1.55	9.04 × 10^−7^
AFLA_060780	conidial hydrophobin dewA	*dewA*			2.58	3.05 × 10^−3^
AFLA_006180	conidial pigment biosynthesis oxidase Arb2/brown2	*arb2*	-	-	1.45	1.10 × 10^−5^
AFLA_006170	polyketide synthetase PksP	*pksP/alb1/wA*	1.83	3.47 × 10^−12^	5.03	9.57 × 10^−90^
AFLA_075640	pigment biosynthesis protein Ayg1	*ayg1*	-	-	2.26	7.39 × 10^−14^

## Data Availability

Data presented in this study are available on request from the corresponding author.

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
