# Peer review of "The Solvent Dimethyl Sulfoxide Affects Physiology, Transcriptome and Secondary Metabolism of Aspergillus flavus"

_jof, 2021, doi:10.3390/jof7121055_

Round 1
Reviewer 1 Report
The solvent Dimethyl Sulfoxide affects physiology, transcriptome and secondary metabolism of Aspergillus flavus
Comments
A very expansive, interesting, useful and easy to read piece of work. However, some few suggestions have been made which will help boost the strength of the paper.
Abstract
Line 15- ‘but this solvent has little biological effects especially on fungi.
Methodology
Line 121- Replace ‘adaptations’ with ‘modifications’
Results
Dose-dependent effect of DMSO on pigmentation
This portion could be enhanced and made more accurate if the color meter was employed and the CIE L a b used to measure the change in color of the Aspergillus cultures. The current method used is somewhat speculating.
Discussion
Line 638- In text reference used NOT appropriate since it is inconsistent with referencing style. Please check and rectify.
Author Response
A very expansive, interesting, useful and easy to read piece of work. However, some few suggestions have been made which will help boost the strength of the paper.
Answer: Authors would like to thank the reviewer for its comments on our work.
Abstract
Line 15- ‘but this solvent has little biological effects especially on fungi.
Answer: Sentence was corrected accordingly
Methodology
Line 121- Replace ‘adaptations’ with ‘modifications’
Answer: Text was corrected as recommended
Results
Dose-dependent effect of DMSO on pigmentation
This portion could be enhanced and made more accurate if the color meter was employed and the CIE L a b used to measure the change in color of the Aspergillus cultures. The current method used is somewhat speculating.
Answer: The reviewer is right and the use of a color meter would have allowed to give a numeric scale to the observed dose-dependent depigmentation. However, since we had to choose only 2 concentrations for the RNAseq experiments, we think that the visual aspect of colonies was sufficiently clear and demonstrative to choose the low and high dose used. A precise evaluation of the color change would have been more interesting if we could do a dose-effect study by RNAseq, which, as reviewer know, is hardly imaginable due to cost.
Discussion
Line 638- In text reference used NOT appropriate since it is inconsistent with referencing style. Please check and rectify.
Answer: Reference was removed and changed by corresponding number. Sorry for that mistake.
Reviewer 2 Report
DMSO is a solvent commonly used in experiments to analyse the antimicrobial effect of different substances. This work shows that the use of DMSO generates important physiological, transcriptomic and metabolic changes that must be taken into account when designing experiments.
The manuscript is excellently presented, with an adequate review of the state of the art, and a strict description of the experimental design and methods used.
The results obtained are of high interest for future research. Finally, the discussion allows us to understand the implications of the results obtained, and to compare them with previous knowledge on the regulation of different pathways of interest for the control of A. flavus. It has been a pleasure for me to review this work.
I only have minor comments and the correction of formatting and grammar issues to consider that the article should be accepted. They are as follows:
Line 58. Candida spp.
Figure 3B bottom. Replace the chinese symbol.
Figure 8A. The colour key (D1/D2) is interchanged. Why is the Q-value not shown in this figure?
Table 1. Indicate the results that belong to D1 and D2 in the table header.
Line 562. (spores/cm2). Superscripts and subindices should be checked throughout the manuscript.
Line 597. What does 5-AC mean?
Line 638. Remove (Umemura et al., 2013).
Lines 659, 665, 668, 668, 702, 704, etc.. VeA, LaeA, WetA, VosA. The format of gene naming needs to be revised.
Line 694. Remove (Calvo et al. 2002)
Line 715. Italicise "A."
Author Response
DMSO is a solvent commonly used in experiments to analyse the antimicrobial effect of different substances. This work shows that the use of DMSO generates important physiological, transcriptomic and metabolic changes that must be taken into account when designing experiments.
The manuscript is excellently presented, with an adequate review of the state of the art, and a strict description of the experimental design and methods used.
The results obtained are of high interest for future research. Finally, the discussion allows us to understand the implications of the results obtained, and to compare them with previous knowledge on the regulation of different pathways of interest for the control of A. flavus. It has been a pleasure for me to review this work.
Answer: Authors would like to thank a lot the reviewer for his comments on our work.
I only have minor comments and the correction of formatting and grammar issues to consider that the article should be accepted. They are as follows:
Line 58. Candida spp.
Answer: The text was corrected accordingly
Figure 3B bottom. Replace the chinese symbol.
Answer: This was corrected in the revised version. The Chinese symbol appeared during pdf conversion of the word file.
Figure 8A. The colour key (D1/D2) is interchanged. Why is the Q-value not shown in this figure?
Answer: Color key was corrected. Thank you.
Regarding Q-value, maybe the reviewer asks about the absence of stars on differentially expressed genes (as done on figure 9). In fact, as it was indicated in the figure caption, “all genes are significantly differentially expressed (q-value <0.05) except those with the note “ns” below”. Since all genes but 3 were differentially expressed, we have chosen this presentation to limit the number of stars that would have make the figure less readable.
To make it clearer, figure caption was changed to:
“Only genes for which q-value >0.05 are indicated as not significantly differentially expressed with “ns” below the bar plot.
We hope it will suit the reviewer.
If the editor think it is necessary, we can modify the figure and put stars under all differentially expressed genes and remove the “ns” under the 3 non-affected genes.
Table 1. Indicate the results that belong to D1 and D2 in the table header.
Answer: Table header was corrected accordingly
Line 562. (spores/cm2). Superscripts and subindices should be checked throughout the manuscript.
Answer: We corrected this mistake and checked the manuscript
Line 597. What does 5-AC mean?
Answer: It corresponds to 5-azacytidine. This was written in full letters in the text.
Line 638. Remove (Umemura et al., 2013).
Answer: Thank you for the remark. Reference was removed and replaced by corresponding number: [65]
Lines 659, 665, 668, 668, 702, 704, etc.. VeA, LaeA, WetA, VosA. The format of gene naming needs to be revised.
Answer: All formats were checked and corrected
Line 694. Remove (Calvo et al. 2002)
Answer: Reference was removed and replaced by corresponding number: [79]
Line 715. Italicise "A."
Answer: Corrected accordingly, sorry for the tipping mistake.